# Feasibility, reproducibility and validity of the 10 meter Shuttle Test in mild to moderately impaired people with stroke

**Harriet Wittink**[1]*, **Tim Blatter**[1], **Jacqueline Outermans**[1], **Mariella Volkers**[2], **Paul Westers**[3], **Olaf Verschuren**[2]

**1** Research Group Lifestyle and Health, Utrecht University of Applied Sciences, Utrecht, The Netherlands, **2** UMC Utrecht Brain Center and Center of Excellence for Rehabilitation Medicine, Utrecht University, Utrecht, The Netherlands, **3** Julius Centre for Health Sciences and Primary Care, University Medical Center Utrecht, Utrecht, The Netherlands

* harriet.wittink@hu.nl

## Abstract

### Background

There currently is no field test available for measuring maximal exercise capacity in people with stroke.

### Objective

To determine the feasibility, reproducibility and validity of the Shuttle Test (ST) to measure exercise capacity in people with stroke.

### Design

Longitudinal study design.

### Setting

Rehabilitation department, day care centres from a nursing home and private practices specialized in neuro rehabilitation.

### Subjects

People with subacute or chronic stroke.

### Interventions

A standardized protocol was used to determine feasibility, reproducibility and validity of the 10-meter Shuttle Test (10mST).

### Main measures

Number of shuttles completed, 1st Ventilatory Threshold (1st VT).

**Data Availability Statement:** All data files are available from the DANS database (DOI: 10.17026/dans-2zq-u4xg).

**Funding:** this project was funded by SIA RAAK Publiek: RAAK.PUB03.015 (Dutch Organisation of Scientific Research: http://regieorgaan-sia.nl). The funders had no role in study design, data collection and analysis, decision to publish or preparation of the manuscript.

**Competing interests:** The authors have declared that no competing interests exist.

## Results

The associations of the number of shuttles completed and cardiopulmonary capacity as measured with a portable gas analyser were r > 0.7, confirming good convergent validity in subacute and chronic people with stroke. Criterion validity, however, indicates it is not a valid test for measuring maximal cardiopulmonary capacity ($VO_{2max}$). Only 60% of participants were able to reach the $1^{st}$VT. Higher cardiopulmonary capacity and a higher total score of the lower extremity Motricity Index contributed significantly to a higher number of shuttles walked (p = 0.001).

## Conclusions

The Shuttle Test may be a safe and useful exercise test for people after stroke, but may not be appropriate for use with people who walk slower than 2 km/h or 0.56 m/s.

## Introduction

Rehabilitation of people with stroke is aimed at gaining as much independence as possible in their own environment. Gaining or maintaining adequate cardiopulmonary capacity is an essential requirement for maintaining physical function [1].

The assessment of maximal cardiopulmonary capacity in people with stroke is more challenging than in healthy subjects because they present with stroke-specific impairments such as muscle weakness, fatigue, poor balance, contractures and spasticity, which can compromise maximal cardiopulmonary exercise test outcome [2, 3]. Given the reported difficulty of measuring $VO_{2max}$ in people with stroke [4], we may need to consider alternatives, for instance, the determination of the first ventilatory threshold ($1^{st}$VT). The $1^{st}$VT represents the intensity limit of prolonged exercise above which a transition to anaerobic metabolism begins [5]. As the intensity of exercise begins to increase, the $1^{st}$VT can be identified at the point where the breathing rate begins to increase. It is recommended as an appropriate target intensity level for the prescription of light to moderate exercise. It has been suggested the $1^{st}$VT may be a more specific measure of cardiopulmonary capacity than peak oxygen uptake in people with stroke [5].

Apart from the challenges of maximal cardiopulmonary exercise testing in people with stroke, it requires expensive equipment and trained personnel, not available to many clinicians. Consequently, there is an urgent need for a valid and reproducible field test to assess at least the $1^{st}$VT and if possible, $VO_{2max}$. In clinical settings field tests of walking ability such as the six-minute walk test (6MWT) are often used. Prior studies have concluded that the 6MWT is not an adequate measure of cardiopulmonary capacity after stroke [6–9], as the number meters walked in the 6MWT only has a moderate correlation with $VO_{2peak}$ as measured in a maximal symptom limited exercise test [10].

The 10 meter Shuttle Test (10mST), as modified for people with Cerebral Palsy [11], may be of more use. For many adults with physical disabilities, the original 20-m shuttle test is not suitable, because the starting speed (8 km/h) and increase (0.5 km/h) every minute are beyond their capabilities. The adapted protocol, with a slower starting speed (2.0 km/h) and smaller speed increases (0.25 km/h), has been found to be safe and to have acceptable reproducibility and convergent validity for children with Cerebral Palsy [11]. In this protocol, the number of

shuttles is the primary outcome, which has been found to be reproducible with an $ICC_{2,1}$ = 0.99 [11] in children with cerebral palsy.

High test-retest reliability ($ICC_{2,1}$ = 0.96) with a standard error of measurement of 6% and high significant correlations ($r_p$ = 0.65, p<0.01) between the 10mST and 6MWT have been found in people with chronic stroke [12]. No studies have investigated the ability of the 10mST to accurately assess cardiopulmonary capacity in people with subacute or chronic stroke. Results may be different in people with subacute stroke compared with people with chronic stroke, as the process of motor recovery may confound reproducibility results.

The primary objective of this study was to investigate the feasibility, reproducibility, criterion and convergent validity of the 10mST to measure cardiopulmonary capacity in people with stroke.

Our secondary objective was to determine if there are differences in the feasibility, reproducibility, criterion and convergent validity of the 10mST during the subacute versus the chronic phase after stroke.

## Methods

### Recruitment and subjects

People with stroke receiving rehabilitation were recruited from March–December 2017 from a rehabilitation centre (subacute participants), or from day therapy centres within residential care facilities or private practices specialized in neuro rehabilitation (chronic participants) in Utrecht, the Netherlands. Subjects were included in the present study if they had a diagnosis of stroke according to the WHO definition [13], were aged over 18 years, able to walk with supervision (Functional Ambulation Categories ≥ 3) and be in the final week prior to discharge from the rehabilitation centre. Subacute stroke was defined as time since stroke between 7 days and 6 months and chronic stroke was defined as time since onset > 6 months [14].

Subjects were excluded if they presented with severe cognitive disorder (Mini Mental State Examination <24 points), severe communicative disorder (Utrecht Communication State < 4 points), if they had contraindications to maximal exercise [15], such as severe cardiovascular disease or if they were a recurrent faller, defined as more than two falls in a six-month period [16]. Falls were defined as "an unexpected event in which the participants come to rest on the ground, floor, or lower level" [17].

The study was approved by the Medical Ethical Board of the Utrecht Medical Center. Written informed consent was obtained from each subject. Subjects were treated in accordance to good clinical practice and the declaration of Helsinki [18].

### Procedure

Data were collected by two experienced exercise physiologists (TB and MV). Both assessors were Automated External Defibrillator (AED)-trained. A physician was always stand-by in the immediate surroundings to act in case of emergency. Prior to the first assessment subjects were asked to perform only light physical activity on the day before testing, and not to exercise or smoke for 2 hours before their test visits as it is known this can influence cardiopulmonary capacity test results.

At the first test visit, pre-participation health screening using the American Heart Association /American College of Sports Medicine Health/Fitness Pre-participation Screening Questionnaire [19] was performed according to guidelines of the American College of Sports Medicine [19]. Following the screening, demographic data were collected on age, sex, lesion type, side of stroke and side of hemiplegia Height and weight were measured using calibrated measurement instruments at the first and second visit, on average 1 week later. Recurrent falls

were defined based on self-report [20]. We measured balance with the Tinetti Performance Oriented Mobility Assessment-Balance (POMA-B) [21]. For the POMA-B the minimum score is 0 and the maximum score 16 (no balance problems). Functional lower extremity strength was measured with the lower extremity Motricity Index [22]. The minimum score is 0 and the maximum score is 100 (normal strength).

The 10mST was conducted according to the protocol by Verschuren et al. [11] which starts at 2 km/h and increases speed by 0.25 km/h every level (one minute) to a maximum speed of 7.5 km/h (23 stages).

Two cones were set to provide a between-cone distance of 10 meters. The 10mST required participants to walk between the 2 markers delineating the course, at a set incremental speed determined by an auditory signal (beep), which was played by a standard CD/MP3 player. Each stage lasted about 1 minute. Participants were instructed to keep walking as long as possible, not to talk during the test and to wait to start walking until the beep had sounded. To ensure that the subjects were safe during the test, a researcher assisted them by walking behind or next to them. No gait belts or other devices were used for support, but patients were allowed to use their own assistive devices for walking. An emergency button was always within reach of the subjects and the therapist.

During the test, subjects wore a mobile gas analysis system and a portable wireless ECG (Custocor Custo Med, Ottobrunn, Germany). The mobile gas analysis system consisted of a facemask, a transmitting unit (containing different oxygen and carbon dioxide gas analysers), and a receiving unit. The transmitting unit with facemask and tubing was attached to the subjects with a harness, and the receiving unit was connected to a laptop computer located within 500 m of the transmitting unit. Metabolic stress test software (Metasoft, Version 5.4) was used to measure ECG, heart rate, breath-by-breath minute ventilation, oxygen consumption ($VO_2$) carbon dioxide production ($VCO_2$), and to calculate the Respiratory Exchange Ratio ($RER = VCO_2/VO_2$). The gas analysis system was fully calibrated immediately before each test as recommend by the manufacturer. The validity and reliability of this system is acceptable [23, 24].

The test finished when the subject was limited by dyspnoea or when the subject was unable to maintain the required speed and failed twice to complete a shuttle before the beep sounded. Subjects were asked not to talk during the test, but to use a hand signal to indicate they wanted to stop the test. After the test subjects were asked to walk back and forth slowly for a few minutes as a cool-down period. For perceived exertion the Borg score was measured (range 6–20) immediately after completing the 10mST test.

Subjects were then asked to rest for 20 minutes prior to engaging in a supramaximal constant work rate validation procedure bout of exercise [25] to confirm subjects had reached maximal effort and the highest shuttle they could achieve during the test. Supramaximal means a workload above the peak workload attained during the 10mST. This validation procedure consisted of a constant work rate test of maximal 3 minutes [26] at one shuttle higher than the highest shuttle reached in the 10mST.

### Outcome measures

The primary variable was the number of shuttles completed.

The secondary variables were:

1. $VO_{2max}$ (ml.kg$^{-1}$.min$^{-1}$), defined as a plateau in $VO_2$ or a Respiratory Exchange Rate (RER) of $\geq 1.05$ for 50–64 year olds and RER $\geq 1.0$ for those 65 years and older for both males and females as recommended by Edvardsen et al. [27]. We considered $VO_{2peak}$ as the average of the last 30 sec of the 10mST.

2. $1^{st}$VT was determined by two independent raters using the ventilatory equivalents method, and the nadir of fraction of oxygen ($P_{ET}O2$) according to Binder et al. [28] When the outcomes were uncertain, the V-slope method [28] was used to verify the $1^{st}$VT. We calculated two-way random Intraclass Correlation Coefficients with absolute agreement for the interrater reliability of $1^{st}$ VT and found it to be $ICC_{2,1} = 0.91$ with a 95% CI .38 – .98.

**Data analyses.**   All analyses were performed with the statistical software package SPSS version 24 for Windows (IBM SPSS Statistics 24). All data were checked with tests of normality using Shapiro Wilks statistics. Percentages were calculated for ordinal data. When normally distributed, means and standard deviations (SD) were calculated for descriptive measures. Medians and ranges were reported for non-normally distributed data. We calculated differences in age, POMA and lower extremity Motricity Index between those who were unable to complete the 10mST (defined as inability to complete the first shuttle in the first test) and those who completed the test using independent samples t-tests or Mann-Whitney U tests as appropriate.

Linear mixed modelling was used to determine whether there were significant differences in number of completed shuttles when correcting for time of testing (test or retest), lower extremity Motricity Index, POMA-B, sex and age as the independent variables. We assumed a random intercept.

Feasibility was assessed by recording complications, such as dizziness, cardiac arrhythmias and/or adverse effects (falls) during or after the ST. In addition, we determined if subjects reached maximal effort during the ST by using a supramaximal constant work rate test.

Reproducibility concerns the degree to which repeated measurements in stable persons (test-retest) provide similar answers. In accordance with Terwee et al. [29] we made a distinction between reliability and agreement.

For reliability we calculated a two way mixed Intraclass Correlation Coefficient ($ICC_{2,1}$) with absolute agreement of interrater reliability for the determination of the $1^{st}$VT, the absolute and relative $VO_{2peak}$, $HR_{peak}$ and $RER_{peak}$, distance walked and number of shuttles for the whole sample, chronic survivors and the subacute survivors [30]. For the interpretation of results, single measure ICC values less than 0.5 are indicative of poor reliability, values between 0.5 and 0.75 indicate moderate reliability, values between 0.75 and 0.9 indicate good reliability, and values greater than 0.90 indicate excellent reliability [30].

To determine the level of agreement, Standard Error of Measurement (SEM) and the smallest detectable change ($SDC_{ind}$) were calculated according to the recommendations of Terwee et al. [29].

For cardiopulmonary criterion validity we determined if the 10mST stresses the cardiopulmonary system sufficiently to attain at least the $1^{st}$VT, or $VO_{2max}$ (see definition above). We considered the test valid for stressing the cardiopulmonary system if the $1^{st}$VT could be determined in 80% of subjects during the 10mST.

For convergent validity we hypothesized that there would be a strong positive correlation ($r \geq 0.75$) between the variables number of shuttles completed and aerobic capacity, measured as $VO_{2peak}$ mL.min$^{-1}$ and $VO_{2peak}$ mL.kg$^{-1}$.min$^{-1}$. Correlations ranging from 0.00–0.25 are considered to indicate no relationship, those from 0.25–0.50 a fair degree of relationship, values of 0.50–0.75 a moderate to good and values above 0.75 a good to excellent relationship [30]. In addition, we performed a forward linear regression analyses with the number of shuttles completed as the dependent variable and the lower extremity Motricity Index score and $VO_{2peak}$ mL.kg$^{-1}$.min$^{-1}$ as the independent variables.

Finally, we analysed whether validity and reproducibility of the 10mST was different between participants in the subacute versus the chronic phase after stroke by using unpaired t-tests and Mann-Whitney U tests as appropriate.

## Results

### Participants

Twenty-six participants were recruited for the study. No potential participants were excluded on the basis of the pre-participation screening. N = 6 (23%) were unable to complete the first shuttle in the first test. Participants who were unable to complete the first shuttle were on average older (mean difference (Standard Error (SE)) 13.6 (6.9) years, 95% CI -0.6; 27.7), had a lower POMA score mean difference (SE) 4.1 (0.7), 95% CI -5.5; -2.7) and a lower lower extremity Motricity Index (mean difference (SE) 28.3 (8.1), 95% CI -45.1; -11.5). These six participants were excluded from the retest.

Twenty participants completed the test twice. Demographic data for the test-retest sample are shown in Table 1.

### Test retest reliability

Linear mixed modelling analysis of the whole sample with the number of completed shuttles as the dependent variables adjusting for sex, age, POMA and lower extremity Motricity Index as the dependent variables showed no significant effect of time of testing (p = 0.06). See Table 2.

### Feasibility

There were no adverse events during or after the 10mST, including dizziness, syncope, fainting, chest pains, cardiac arrhythmias or (near) falls while performing the 10mST. Most (n = 17 (85%)) of the participants stopped due to motor control and/or coordination problems that resulted in the inability to walk the required speed. Three (15%) participants reported stopping because they were out of breath. Median (range) Borg scores were 15 (7–17) and 16.5 (7–18) at the test and retest, respectively. The test was viable in all settings (private practice, rehabilitation and day care).

Mean (SD) minutes walked in the supramaximal test was 1.2 (0.6) minutes. No participants were able to complete the 3 minute supramaximal test, suggesting they had reached their maximal capacity during the ST.

### Reproducibility

There were no statistically significant differences between the test and retest for all measures reported in Table 3.

In Table 3 the results of the reproducibility of the Shuttle Test are presented with the single measure $ICC_{2,1 \text{ agreement}}$, SEM and SDC measures for the whole sample. $VO_{2peak}$ $L.min^{-1}$, $VO_{2peak}$ $mL.kg^{-1}.min^{-1}$, number of shuttles and meters walked had ICCs > 0.90 and were considered excellent. For the reproducibility measures for the chronic people with stroke, see S1 Appendix and for the reproducibility measures for the subacute group, see S2 Appendix.

### Validity

For criterion validity we determined if the ST stresses the cardiopulmonary system sufficiently to attain at least the 1st VT or $VO_{2max}$ in at least 80% of participating participants. Ten (50%) out of the 20 participants attained 1stVT in the first test. The same ten participants and an additional two participants (60%) attained a 1stVT in the retest. One participant aged 79 years

**Table 1. Demographic variables of the test-retest participants (n = 20).**

| | Total (n = 20) | Subacute (n = 7) | Chronic (n = 13) | p-value |
|---|---|---|---|---|
| Male (n (%)) | 9 (45) | 4 (57) | 5 (39) | 0.42*** |
| Age (Years, mean(SD)) | 60 (16) | 46 (15) | 67 (12) | **0.005**\* |
| Height (cm, mean(SD)) | 171 (11) | 170 (10) | 171 (11) | 0.87* |
| Weight (kg, mean(SD)) | 78.1 (17.1) | 69.4 (15.0) | 82.7 (16.9) | 0.10* |
| Duration stroke (months, mean (SD)) | 62.3 (78.0) | 7.2 (2.0) | 87.8 (83.1) | **0.004**\* |
| Hemiplegic side (n (%)) | | | | 0.62*** |
| Left | 10 (50) | 3 (43) | 7 (53) | |
| Right | 6 (30) | 3 (43) | 3 (21) | |
| Both | 1 (5) | | 1 (8) | |
| None | 1 (5) | | 1 (8) | |
| Missing | 2 (10) | 1 (14) | 1 (8) | |
| Type of CVA (n (%)) | | | | 0.34*** |
| Ischaemic | 14 (70) | 4 (57) | 10 (77) | |
| Haemorrhagic | 5 (25) | 3 (43) | 2 (15) | |
| Unknown | 1 (5) | | 1 (8) | |
| FAC (n (%)) | | | | 0.48*** |
| FAC 4 | 4 (20) | 2 (29) | 2 (15) | |
| FAC 5 | 16 (80) | 5 (71) | 11 (85) | |
| Motricity Index lower extremity (median (range)) | 91 (44–100) | 100 (89–100) | 85.0 (44–100) | **0.006**\** |
| POMA (median (range)) | 16 (13–16) | 16 (14–16) | 16 (13–16) | 0.94** |
| Beta-blockers (n (%)) | 3 (15) | 2 (29) | 1 (8) | 0.33*** |
| Assistive devices indoors (n (%)) | | | | 0.30*** |
| None | 18 (90) | 6 (86) | 12 (92) | |
| Cane | 1 (5) | 1 (14) | | |
| Walker / cane | 1 (5) | | 1 (8) | |
| Assistive devices outdoors (n (%)) | | | | 0.58*** |
| None | 15 (75) | 6 (87) | 9 (69) | |
| Cane | 2 (10) | 1 (14) | 1 (8) | |
| (rolling) walker | 2 (10) | | 2 (16) | |
| Walker/ cane | 1 (5) | | 1 (8) | |

*Independent t-test,

**Mann-Whitney U test,

***Chi square.

attained a RER = 1.04 in the test and one subject aged 65 years attained a RER = 1.0 in the retest, suggesting that by our criteria they reached $VO_{2max}$.

**Table 2. Test-retest reliability of the number of shuttles completed.**

| Variable | Estimate | t | Sig. | 95% Confidence Interval | |
|---|---|---|---|---|---|
| | | | | Lower Bound | Upper Bound |
| Intercept | -17.69 | -1.18 | 0.26 | -49.71 | 14.34 |
| Time of testing | 0.48 | 1.99 | 0.06 | -0.03 | 0.98 |
| Sex | 3.44 | 1.90 | 0.08 | -0.41 | 7.29 |
| POMA | 1.66 | 1.84 | 0.09 | -0.26 | 3.58 |
| lower extremity Motricity Index | 0.110 | 1.80 | 0.09 | -0.02 | 0.23 |
| Age | -0.15 | -2.63 | 0.02 | -0.27 | -0.03 |

**Table 3. Reproducibility Shuttle Test whole sample.**

| Mean (SD) | Test (n = 20) | Retest (n = 20) | Single measure $ICC_{1,2}$ agreement 95% CI | SEM | $SDC_{ind}$ | $SDC_{group}$ |
|---|---|---|---|---|---|---|
| $VO_{2peak}$ (L.min$^{-1}$) | 1.67 (0.7) | 1.6 (0.7) | 0.9 (0.8; 1.0) | 0.2 | 0.5 | 0.1 |
| $VO_{2peak}$ (mL.kg$^{-1}$.min$^{-1}$) | 20.3 (7.5) | 21.05 (7.7) | 0.9 (0.8; 1.0) | 2.4 | 6.7 | 1.9 |
| $HR_{peak}$ (beats per minute) | 125 (30) | 126 (29) | 0.9 (0.8; 1.0) | 9.7 | 26.8 | 7.4 |
| $RER_{peak}$ | 0.9 (0.1) | 0.9 (0.1) | 0.7 (0.4; 0.9) | 0.0 | 0.1 | 0.0 |
| distance walked (meters) | 593 (374) | 632 (374) | 0.97 (0.92; 0.99) | 64.8 | 179.7 | 49.8 |
| Number of shuttles completed | 9.9 (5.3) | 10.4 (5.5) | 1.0 (0.9; 1.0) | 0.8 | 2.1 | 0.6 |
| 1$^{st}$VT (L.min$^{-1}$) | 1.7 (.4) | 1.7 (.5) | 0.8 (0.5; 1.0) | 0.2 | 0.5 | 0.2 |
| 1$^{st}$VT (mL.kg$^{-1}$.min$^{-1}$) | 20.5 (5.0) | 20.9 (3.9) | 0.7 (0.1; 0.9) | 2.6 | 7.1 | 2.0 |

$HR_{peak}$ = peak Heart Rate, $RER_{peak}$ - = peak Respiratory Exchange Rate, SEM = Standard error of Measurement, SDC = Smallest Detectable Change.

Two participants attained the 1$^{st}$VT at the end of both test and retest and three additional participants attained the 1$^{st}$VT at the end of the retest. There were no significant differences between test and retest measures of 1$^{st}$VT.

The mean difference (SE) in the lower extremity Motricity Index between participants who attained a 1$^{st}$VT and those who did not was 10.4 (6.8) 95%CI -24.6; 3.8. In the retest the mean difference (SE) in the lower extremity Motricity Index was 14.5 (6.5), 95% CI -28.2; -0.9. In both instances the lower extremity Motricity Index was lower for those who did not attain a 1$^{st}$VT, but this was only statistically significantly different in the retest.

For convergent validity we found 2-sided Pearson correlations between number of shuttles completed and cardiovascular values were significant ($p < 0.01$) for $VO_{2peak}$ L.min$^{-1}$ r = 0.7 (test) and r = 0.8 (retest) and for $VO_{2peak}$ mL.kg$^{-1}$.min$^{-1}$ r = 0.8 (test) and r = 0.9 (retest) (see Fig 1).

The total lower extremity Motricity Index had a significant association ($p < 0.01$) of r = 0.6 with the number of shuttles completed for the test and retest, respectively. There was no significant association of the number of shuttles with sex, body height or weight.

Forward multivariate analysis with the number shuttles completed at the first test as the dependent variable and lower extremity Motricity Index and $VO_{2peak}$ ml.kg$^{-1}$.min$^{-1}$ as independent variables showed an adjusted $R^2$ of 0.6 in the final model with $VO_{2peak}$ mL.kg$^{-1}$.min$^{-1}$ ($p<0.001$).

At the retest, the adjusted $R^2$ was 0.76 with lower extremity Motricity Index (p = 0.05) and $VO_{2peak}$ mL.kg$^{-1}$.min$^{-1}$ (p = 0.001) contributing significantly to the model (see Table 4).

## Differences in people with stroke in the subacute versus chronic phase

Seven in-patient rehabilitation (subacute) participants, 4 males and 3 females, were included in the sample. Independent t-testing between the subacute sample and the sample with chronic stroke revealed the subacute sample was significantly younger (mean difference (SE) = 20 (6) years, 95% 7.1;33.0) Their median lower extremity Motricity Index was significantly higher (p = 0.007). The mean difference (SE) in number of meters walked (327.2 (162.9), 95% CI -669.5; 15.2) did not significantly differ between groups in the test, but was significantly higher in the retest 378.6 (175.90 meters, 95% CI -748.2; -8.9. $VO_{2peak}$ L.min$^{-1}$ did not differ between the groups. In contrast, the mean difference (SE) $VO_{2peak}$ ml.kg$^{-1}$.min$^{-1}$ was significantly higher in both test 9.0 (3.6) ml.kg$^{-1}$.min$^{-1}$, 95% CI -17.4;-0.5 and retest 9.2 (3.7) ml.kg$^{-1}$.min$^{-1}$, 95% CI -17.6; -0.7 for the subacute sample. For details, see S3 Appendix.

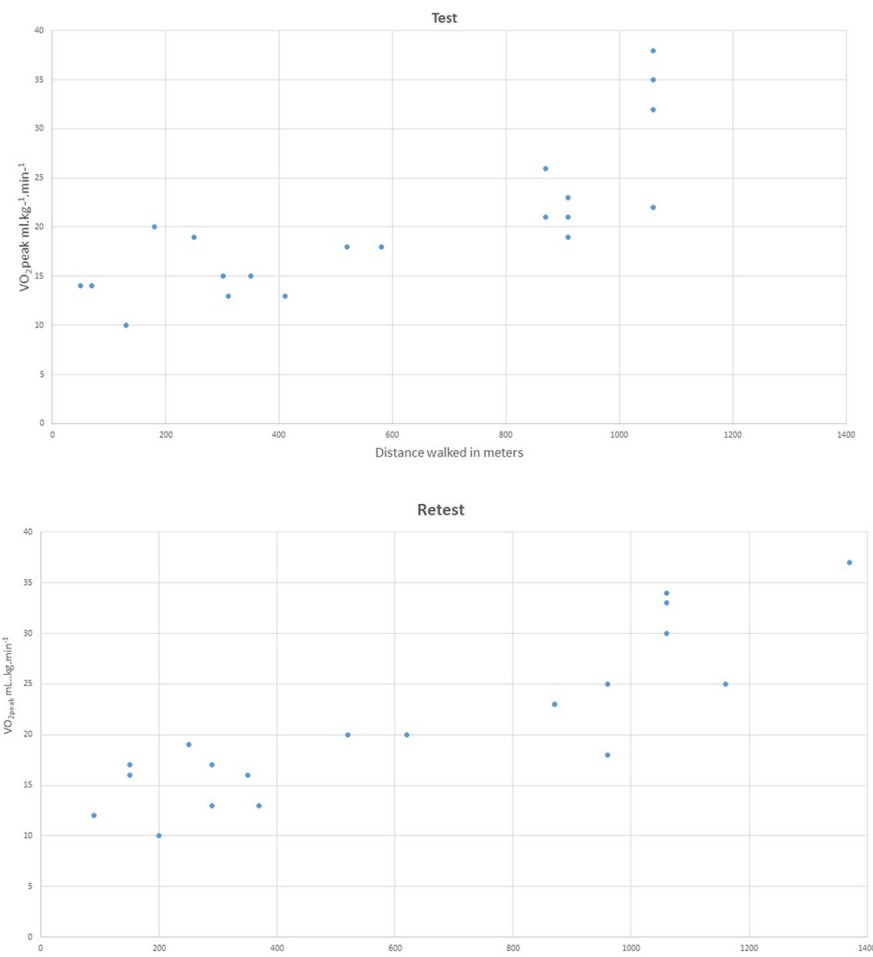

**Fig 1. Scatterplot distance walked in meters and VO$_{2peak}$ mL.kg.min$^{-1}$ test and retest.**

## Discussion

This study provides insight into the cardiopulmonary response to the 10mST in people with stroke both earlier (within 3 months) and later ($\geq$ 6 months post-stroke). None of the participants were able to complete the supramaximal test, suggesting they reached maximal exertion in the 10mST. Therefore, the 10mST can be considered a symptom limited maximal exercise test for people with stroke. The metabolic demand of the 10mST varied widely as the primary limiting factor was motor performance for most participants.

The 10mST is a safe, feasible and reproducible test for measuring symptom limited maximal exercise capacity with good convergent validity people with stroke. Criterion validity, however, indicates it is not a valid test for measuring maximal cardiopulmonary capacity. Eighty-five percent of participants in the test-retest sample stopped due to motor and/or coordination problems. By our criteria, only two participants (5%) achieved a VO$_{2max}$ in the test or retest; both of these participants had minimal motor impairments. A 1$^{st}$VT could only be determined in 50% of participants during the first test and in 60% during the second test.

For convergent validity we found a good to excellent relationship between the number of shuttles completed and cardiopulmonary capacity. This was confirmed in the multiple regression

**Table 4. Coefficients for number of shuttles (dependent variable) at retest.**

| Variables | | Standardized Coefficients | t | Sig. | 95% Confidence Interval for B | |
|---|---|---|---|---|---|---|
| | | Beta | | | Lower Bound | Upper Bound |
| 1 | (Constant) | | -1.237 | 0.23 | -6.471 | 1.674 |
| | RelVO$_{2peak}$T2 | 0.855 | 7.006 | 0.001 | 0.426 | 0.790 |
| 2 | (Constant) | | -2.510 | 0.02 | -15.674 | -1.356 |
| | RelVO$_{2peak}$T2 | 0.743 | 6.011 | 0.001 | 0.343 | 0.714 |
| | Lower extremity Motricity Index | 0.262 | 2.115 | 0.05 | 0.001 | 0.182 |

analysis where VO$_{2peak}$ mL.kg$^{-1}$.min$^{-1}$ made a significant contribution to the number of shuttles completed. Participants with higher cardiopulmonary capacity were able to walk more shuttles.

Reproducibility was consistent with the results reported by van Bloemendaal et al. [12] who reported similar ICC$_{2,1\ agreement}$ scores, but a higher standard error of measurement (109 m, compared with 65 m in our study. Similar to Bongers et al. [31] in our study all retest measures were slightly higher, which suggests a learning effect. This is in contrast to Bloemendaal et al. [12] who did not find a learning effect. Nevertheless, a habituation session prior to the 10mST is recommended.

The finding that those with lower lower extremity Motricity Index (i.e. less strength) were less likely to reach a 1$^{st}$VT, matches the participants' reporting stopping due to motor / coordination problems. This is in contrast to Marzolini et al. [32] who reported 76% of participants had a discernible 1$^{st}$VT during a maximal exercise test on a recumbent bicycle, and did not differ significantly from those who did not reach a 1$^{st}$VT by lower limb motor impairment and balance score.

## Study strengths and weaknesses

This is the first study to demonstrate reproducibility of a field-based cardiopulmonary capacity / fitness test in people with stroke [4]. However, our sample size was small and six participants (23%) of the initial sample were unable to perform the first level of the 10mST. Apart from being very slow walkers (< 2 km/h or 0.56 m/s), they also had significantly less lower extremity motor strength and balance. The 10mST may therefore not be a useful test for very slow walkers (those walking 2 km/h or slower) or for people with severe balance problems (those scoring 11 or less on the POMA).

Prior research showed balance, measured by the POMA, has a confounding effect on distance walked in the 6MWT [33], but we did not find this effect in this study. As we excluded participants with a history of falls, potentially excluding those with poorer balance, we may have limited this confounding effect. This could also have positively affected the number of shuttles completed. We did not measure the distance walked, but estimated it from the number of shuttles walked. This could have resulted in overly optimistic reproducibility measures for the distance walked.

## Implications

Walk tests, such as the commonly used 6-minute walk test (6MWT), have been designed and used as surrogate measures of cardiorespiratory capacity in other populations, but associations between distance walked and cardiopulmonary capacity are low to moderate in people with stroke [10]. Correlations between the 10mST and cardiopulmonary capacity were higher (r>0.7) than those reported for the 6MWT (mean r = 0.5, 95% CI 0.4–0.6) [10], suggesting the 10mST may be a better measure for assessing cardiopulmonary capacity than the 6MWT.

In addition, the 10mST is a maximal symptom limited exercise test and may be useful as a screener for exercise tolerance and the potential adverse effects of exercise.

Further studies should examine the responsiveness and minimally important change in this population to determine if the test can be used as an outcome measure.

## Clinical messages

- The 10mST is a maximal symptom limited exercise test

- The 10mST is a safe, feasible and reproducible test in people with subacute and chronic stroke

- The 10mST has a good to excellent relationship with cardiopulmonary capacity

## Supporting information

**S1 Appendix. Reproducibility 10mST chronic people with stroke.**
(DOCX)

**S2 Appendix. Reproducibility 10mST subacute people with stroke.** $HR_{peak}$ = peak Heart Rate, $RER_{peak}$ - = peak Respiratory Exchange Rate, SEM = Standard error of Measurement, SDC = Smallest Detectable Change.
(DOCX)

**S3 Appendix. Differences in test results between persons with stroke in the subacute and chronic phase.**
(DOCX)

## Acknowledgments

The authors wish to thank the participating physical therapists/ kinetic therapists: Evelien Bruggemann; Jolien Netjes; Maud Eeuwen; Manoek Ticheler; Marloes Everaers; Judith Meijer-ink; Marielle Wittekamp; Jonneke Kroes and the people with stroke who participated in this study.

## Author Contributions

**Conceptualization:** Harriet Wittink, Tim Blatter, Jacqueline Outermans, Olaf Verschuren.

**Data curation:** Harriet Wittink.

**Formal analysis:** Harriet Wittink, Paul Westers.

**Funding acquisition:** Harriet Wittink, Olaf Verschuren.

**Investigation:** Harriet Wittink, Tim Blatter, Mariella Volkers.

**Methodology:** Harriet Wittink, Jacqueline Outermans, Paul Westers.

**Project administration:** Harriet Wittink, Tim Blatter, Jacqueline Outermans, Mariella Volkers, Olaf Verschuren.

**Resources:** Harriet Wittink.

**Supervision:** Harriet Wittink, Jacqueline Outermans, Olaf Verschuren.

**Validation:** Harriet Wittink, Mariella Volkers, Olaf Verschuren.

**Writing – original draft:** Harriet Wittink.

**Writing – review & editing:** Harriet Wittink, Tim Blatter, Jacqueline Outermans, Mariella Volkers, Paul Westers, Olaf Verschuren.

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
