## [Decision Letter · Decision Letter 0]

18 Mar 2020

PONE-D-19-34382

Feasibility, reproducibility and validity of the 10 meter Shuttle Test in stroke survivors

PLOS ONE

Dear Prof. dr. Wittink,

Thank you for submitting your manuscript to PLOS ONE. After careful consideration, we feel that it has merit but does not fully meet PLOS ONE’s publication criteria as it currently stands. Therefore, we invite you to submit a revised version of the manuscript that addresses the points raised during the review process.

We would appreciate receiving your revised manuscript by May 02 2020 11:59PM. To enhance the reproducibility of your results, we recommend that if applicable you deposit your laboratory protocols in protocols.io, where a protocol can be assigned its own identifier (DOI) such that it can be cited independently in the future. For instructions see: http://journals.plos.org/plosone/s/submission-guidelines#loc-laboratory-protocols

We look forward to receiving your revised manuscript.

Kind regards,

Gerson Cipriano Jr., PT, MsC, Ph.D.

Academic Editor

PLOS ONE

Additional Editor Comments (if provided):

Thank you for submitting your paper to PlosOne. We have now completed our review.

The purpose of this research report was to evaluate the psychometric properties of an adapted version of the Incremental Shuttle Walk Test in patients with subacute and chronic stroke.

The reviewers and I agree that your manuscript could potentially make an impactful contribution to the rehabilitation literature, although it requires some major review.

Please complete the following revisions:

Journal Requirements:

Reviewers' comments:

Reviewer's Responses to Questions

**Comments to the Author**

1. Is the manuscript technically sound, and do the data support the conclusions?

Reviewer #1: Yes

Reviewer #2: Yes

2. Has the statistical analysis been performed appropriately and rigorously? 

Reviewer #1: Yes

Reviewer #2: Yes

3. Have the authors made all data underlying the findings in their manuscript fully available?

Reviewer #1: No

Reviewer #2: Yes

4. Is the manuscript presented in an intelligible fashion and written in standard English?

Reviewer #1: Yes

Reviewer #2: Yes

5. Review Comments to the Author

Reviewer #1: The current study aimed at assessing the psychometric properties of an adapted version of the Incremental Shuttle Walk Test in stroke survivors. The authors also examined whether differences in psychometric quality of the presented instrument existed in individuals grouped by length of time after stroke onset. Although this is a well-written study which adds to current evidence, I have a few concerns about the manuscript that I would like the authors to address. Please see the below comments and questions.

1. While the authors are aware of the study aspects that have potentially limited the generalization of findings, they fail to adequately present the study population in both the title of the manuscript and the conclusion. It should be made clear to readers that the reported findings are only valid in “highly functioning stroke survivors” or another similar, more specific possibility.

2. Two independent raters determined measures of 1st VT. Yet, no interrater reliability data have been reported. Please analyze and provide the degree of agreement among the examiners.

3. The authors use the term “reproducibility” throughout the text when, in fact, this should be named “repeatability” or even "test-retest reliability". There is a difference between reproducibility and repeatability.

4. There are two types of construct validity but the authors only report on one of them: convergent validity. Please use this term instead.

5. Still on the convergent validity analysis: please provide scatter plots to illustrate the most significant relationships observed.

6. Bland-Altman plots should have been provided and used to determine if any systematic differences across the range of values occurred between the two testing sessions.

Minor edits and questions:

7. Please define “close to completion” (line 112, p. 6)

8. Please specify how the “recurrent faller” info was obtained (i.e. medical records, self-reported…)

9. Was a gait belt or any other device used for support during the tests?

10. Why was a 20-minute recovery period chosen? I wonder whether this was enough time for them to fully recover.

11. Were individuals asked to withdraw medications for the tests?

12. Please clarify “time of testing”. What do you mean? Daytime vs. nighttime?

13. Reliability info was provided for the entire sample and for the subacute group, but not for the chronic group specifically.

Reviewer #2: This paper concerned the use of the shuttle test to assess the cardiopulmonary fitness of patients with subacute and chronic stroke. Evaluating the psychometrics of the shuttle test in this population was the main objective. Overall, this is a technically sound submission that addresses a specific gap in existing literature regarding the use of this specific outcome measure in this specific patient population. To the extent of the knowledge of this reviewer it is original work that has not previously been published elsewhere.

Areas of Primary Concern:

-This study used as exclusion criteria patients with more significant balance dysfunction as defined by a history of “recurrent falls” (was this assessed via self report?), and then a total of 6 out of 26 recruited subjects did not complete the entire study due to the inability to complete the first stage of the test, thus reducing the overall sample population assessed to those with subacute/chronic stroke and higher levels of walking ability. This was discussed within the body paper, yet the title of the paper could be misleading in that it seems to indicate that the study population was all “stroke survivors”. Please consider re-writing the title to indicate the specific nature of the study population.

-The authors use the term reproducibility throughout the paper regarding the stability of results with administration of the outcome measure in the same patients on two different occasions separate by a time period, but perhaps what they mean is test-retest reliability or repeatability?

-It is not reported whether there was any assessment of tone in this study, which is a variable that may be likely to have an effect or interaction with the outcomes and would have been interesting to include in the baseline assessments of the patient population

-It was reported in the data tables how many patients used assistive devices out of the overall sample, but it was not reported whether these patients used their assistive devices during the performance of the shuttle test

-Line 76: Please consider re-wording this statement: The wording regarding “cardiac vs non-cardiac” exercise limitation should probably be rephrased to state that inability to reach the 1st VT only helps you distinguish whether the patient is limited due to their capacity to sustain adequate aerobic metabolism to support a given exercise test, versus being limited by another reason (in the case of this patient population, motor control/neuromuscular function). It is not sensitive enough to distinguish between the dysfunction in aerobic capacity that could be attributed to a particular organ system (i.e. cardiac vs pulmonary) as this sentence seems to indicate.

-Please review the total summation of the numbers of people using assistive device in Table 1 starting at line 609 “Assistive devices indoors”, the totals seem to be off by 1

-Would like to have seen a 6MWT performed so that those results could be related to this study population, and since the 6MWT is the only other highly commonly used outcome measure in clinical practice. Especially since this study population was so filtered to include higher-functioning individuals with better balance. Perhaps the ST only looks like a more reliable measure because the study population was filtered in this way? We are unable to compare these results to other studies that have looked at 6MWT.

6. PLOS authors have the option to publish the peer review history of their article (what does this mean?). If published, this will include your full peer review and any attached files.

Reviewer #1: No

Reviewer #2: Yes: Dr. Brady Anderson

---

## [Author Response · Author response to Decision Letter 0]

22 Apr 2020

Please see letter"Response to reviewers"

---

## [Editor Report · Decision Letter 1]

9 Jun 2020

PONE-D-19-34382R1

Feasibility, reproducibility and validity of the 10 meter Shuttle Test in mild to moderately impaired stroke survivors

PLOS ONE

Dear Dr. Wittink,

Thank you for submitting your manuscript to PLOS ONE. After careful consideration, we feel that it has merit but does not fully meet PLOS ONE’s publication criteria as it currently stands. Therefore, we invite you to submit a revised version of the manuscript that addresses the points raised during the review process.

I realise you have been waiting a long time for the decision. I have only just taken it over as Academic Editor. I feel it has real merit and is an important paper to have published. I have made quite extensive suggestions for edits (see attached word document). The purpose of the revisions is to the make the paper easier to read, and therefore more likely to be widely read and impactful.

I encourage you to complete these as soon as you can and I will do my best to expedite the process as soon as I receive them.

Congratulations on an important piece of work

A rebuttal letter that responds to each point raised by the academic editor and reviewer(s). You should upload this letter as a separate file labeled 'Response to Reviewers'. (Please only refer to major changes or areas you disagree with. There is no need to list changes you have simply accepted.)A marked-up copy of your manuscript that highlights changes made to the original version. You should upload this as a separate file labeled 'Revised Manuscript with Track Changes'.An unmarked version of your revised paper without tracked changes. You should upload this as a separate file labeled 'Manuscript'.

We look forward to receiving your revised manuscript.

Kind regards,

Coralie English, PhD

Academic Editor

PLOS ONE

---

## [Author Response · Author response to Decision Letter 1]

1 Jul 2020

Please see our response to reviewers letter.

---

## [Editor Report · Decision Letter 2]

14 Jul 2020

PONE-D-19-34382R2

Feasibility, reproducibility and validity of the 10 meter Shuttle Test in mild to moderately impaired people with stroke

PLOS ONE

Dear Dr. Wittink,

Thank you for submitting your manuscript to PLOS ONE. After careful consideration, we feel that it has merit but does not fully meet PLOS ONE’s publication criteria as it currently stands. Therefore, we invite you to submit a revised version of the manuscript that addresses the points raised during the review process.

We look forward to receiving your revised manuscript.

Kind regards,

Coralie English

Academic Editor

PLOS ONE

Journal Requirements:

Additional Editor Comments (if provided):

Thank you for your resubmission. Pease complete the few minor changes as outlined in the attached document
---

## [Author Response · Author response to Decision Letter 2]

25 Aug 2020

please see letter "response to reviewers".

---

## [Editor Report · Decision Letter 3]

2 Sep 2020

Feasibility, reproducibility and validity of the 10 meter Shuttle Test in mild to moderately impaired people with stroke

PONE-D-19-34382R3

Dear Dr. Wittink,

We’re pleased to inform you that your manuscript has been judged scientifically suitable for publication and will be formally accepted for publication once it meets all outstanding technical requirements.

Kind regards,

Coralie English, PhD

Academic Editor

PLOS ONE
---

## [Editor Report · Acceptance letter]

16 Oct 2020

PONE-D-19-34382R3 

Feasibility, reproducibility and validity of the 10 meter Shuttle Test in mild to moderately impaired people with stroke. 

Dear Dr. Wittink:

I'm pleased to inform you that your manuscript has been deemed suitable for publication in PLOS ONE. Congratulations! Your manuscript is now with our production department. 

Kind regards, 

on behalf of

Professor Coralie English 

Academic Editor

PLOS ONE